# Home-based Intervention with Semaglutide Treatment of Neuroleptic-Related Prediabetes (HISTORI): protocol describing a prospective, randomised, placebo controlled and double-blinded multicentre trial

Ashok Ainkaran Ganeshalingam [1,2,3] Nicolai Gundtoft Uhrenholt,[4,5] Sidse Arnfred,[6] Peter Haulund Gæde,[7] Niels Bilenberg,[5,8] Jan Frystyk[1]

For numbered affiliations see end of article.

**Correspondence to**
Dr Ashok Ainkaran Ganeshalingam; ashok.ganeshalingam@rsyd.dk

## ABSTRACT

**Introduction** Subjects with schizophrenia have a 2–3 fold higher mortality rate than the general population and a reduced life expectancy of 10–20 years. Approximately one-third of this excess mortality has been attributed to obesity-related type 2 diabetes (T2D) and to cardiovascular disease. Glucagon-like peptide-1 (GLP-1) analogues increase satiety and delay gastric emptying, thereby reducing food intake and weight. GLP-1 analogues also exert beneficial effects on cardiovascular outcomes in high-risk patients with T2D.

Our aim is to investigate whether 30 weeks add-on treatment with the GLP-1 analogue semaglutide can reduce HbA1c sufficiently to reverse pre-diabetes and the metabolic syndrome in overweight schizophrenic patients.

**Methods and analysis** We will perform a 30 week, two-armed, multicentre, superiority, double-blinded, randomised trial investigating the effect of weekly injections of semaglutide versus placebo in mental health facilities in Region of Southern Denmark and Region of Zealand, Denmark. In total, 154 adults with schizophrenia spectrum disease, aged 18–60 years treated with second generation antipsychotic treatment, HbA1c 39–47 mmol/mol and body mass index >27 kg/m$^2$ will be randomised to injections of 1.0 mg semaglutide or placebo. The primary outcome is changes in HbA1c. Secondary outcomes encompass metabolic measures, psychotic symptoms and quality of life. Exploratory outcomes encompass insulin sensitivity, cardiovascular risk profile, medication adherence, general well-being and physical activity.

**Ethics and dissemination** This study will be carried out in accordance with the Declaration of Helsinki and Good Clinical Practice guidelines. This research has obtained approval from both the Danish Medicines Agency and The Regional Committees on Health Research Ethics for Southern Denmark.

**Trial registration number** NCT05193578
European Clinical Trials Database Number (EudraCT)

## STRENGTHS AND LIMITATIONS OF THIS STUDY

⇒ A 30-week trial designed with feedback from patients and relatives, conducted as a double-blinded, randomised, parallel-group and placebo-controlled study, adhering to good clinical practice standards and monitored across multiple centres.

⇒ The integration of endocrinology and psychiatry in our study holds significant value as it provides a holistic understanding of the intricate interplay between hormonal regulation and mental health, offering potential insights into innovative approaches for comprehensive patient care.

⇒ This study employs a unique home-based intervention approach, facilitating data collection within participants' natural environments, enabling individuals experiencing severe symptoms of schizophrenia to engage in the study.

⇒ Despite the significance of the study, it is important to note that the 30-week trial duration may impose limitations on fully exploring the comprehensive potential of glucagon-like peptide-1 receptor agonists.

2020-004374-22, Regional Ethical Committee number S-20200182.

## BACKGROUND AND SIGNIFICANCE OF HISTORI

The life expectancy of subjects with schizophrenia spectrum disorders (SZD) (hereafter schizophrenia) is reduced by 10–20 years when compared with the general population.[1–3] The excess mortality is partly explained by a 2–3 fold increase in cardiovascular disease (CVD).[4 5] A nationwide register study from Sweden including more than 46 000 subjects with schizophrenia showed that CVD was the leading cause of death,

BMJ

causing more deaths than suicide and that death from CVD occurred 10 years earlier in schizophrenic subjects than in the general population.[5]

Another factor predisposing to the reduced life expectancy is type 2 diabetes (T2D), which develops in 10%–15% of all patients with schizophrenia.[6] Compared with the general population, patients with schizophrenia are more likely to have diabetes, hypertension, dyslipidaemia, to be overweight and to smoke.[7] Indeed, one-third of patients with schizophrenia have been reported to suffer from the metabolic syndrome (MetS).[8]

Antipsychotic treatment is vital to reduce schizophrenia-related symptoms[9] and treatment with second generation anti-psychotic (SGA) drugs has demonstrated beneficial effects on mortality, suicide rates and hospitalisations.[10] However, SGA treatment comes with unfortunate side effects, which typically include increased risks of developing obesity, MetS and pre-diabetes.[11 12] Furthermore, SGA treatment per se leads to impaired glucose tolerance, diabetes, hyperlipidaemia and CVD.[7 13]

Gaining weight is associated with a reduced quality of life (QoL) in people with schizophrenia.[14-19] In addition, weight gain and MetS are among the most frequently reported reasons for discontinuation of treatment in this group.[20] Indeed, in a cohort of 304 subjects with schizophrenia, 94% reported distress over body weight and those who were obese appeared to be more than twice as likely to be non-adherent compared with those with a body mass index (BMI) below $25 \, kg/m^2$.[21] Therefore, it may be speculated that weight-reducing treatments may lead to an increased adherence to SGA treatment. However, to the best of our knowledge, the association between a glucagon-like peptide 1 (GLP-1) agonist-induced weight reduction, the severity of psychotic symptoms and the ability to adhere to SGA treatment has never been investigated in a randomised clinical trial (RCT).

Lifestyle and environmental factors such as smoking, poor diet and an insufficient capacity to exercise contribute to the risk of CVD.[13] Behavioural lifestyle interventions aiming at improving diet and physical activity have indeed shown short-term beneficial effects, but long-term effects diminish over time compared with standard care.[22 23] When lifestyle programmes fail to reduce the increased risk of CVD and T2D, metformin is standard treatment according to NICE guidelines (NICE 2012). In a meta-analysis of 19 RCTs (N=1279), the addition of metformin to ongoing antipsychotic treatment for an average of 3–4 months in subjects with severe mental illness reduced body weight but only by 0.61 kg.[24] Therefore, alternative weight reducing drugs are required, for example, GLP-1 analogues.

The potential of GLP-1 analogues to improve the metabolic profile of patients with schizophrenia was recently successfully tested. In a randomised controlled trial, including 97 patients (placebo/active comparator: 50/47) with schizophrenia, pre-diabetes and a BMI $\geq 27 \, kg/m^2$, 16 weeks of liraglutide (1.8 mg Victoza daily) reduced body weight, waist circumference and abdominal fat mass, LDL cholesterol, glucose levels and insulin resistance, without any worsening in the psychiatric disorder.[25] The beneficial effect of GLP-1 analogue treatment in SGA-treated patients was later reconfirmed in a 2018 meta-analysis which, however, only included two additional and smaller trials of Exanatide (Bydureon), and only one of these studies was placebo controlled.[26] Nevertheless, the review concluded that GLP-1 analogue treatment is effective and tolerable for SGA-related weight gain, but that more studies are required.[27 28]

Today, liraglutide has been replaced by semaglutide, which is more efficient as a weight losing agent in obese subjects.[29] Furthermore, semaglutide is easier to administer than liraglutide as semaglutide requires weekly and not daily injections. This may be important as it opens for the possibility of administering semaglutide by mobile nurses performing weekly house calls, thereby improving adherence to treatment, which is known to be reduced in patients with schizophrenia.[30] Importantly, in non-diabetic obese subjects, semaglutide is able to improve various cardiometabolic risk factors[31] and to prevent new cardiovascular events (death from cardiovascular causes as well as non-fatal myocardial infarction and stroke) in subjects with pre-existing CVD.[32] Therefore, semaglutide appears as a suitable treatment for obese schizophrenic patients.

### Aim and hypothesis

The aim of this study is to perform a randomised placebo-controlled double-blinded trial of semaglutide 1.0 mg in comparison to placebo in patients with SZD, that is, either schizophrenia, schizotypal disorder or schizoaffective disorder and comorbid obesity and pre-diabetes. (The study was planned and initiated prior to the release of semaglutide 2.4 mg; Wegovy®).

We hypothesise that administration of semaglutide at a dose of 1.0 mg administered subcutaneously once a week for 30 weeks improves HbA1c (primary outcome) in SGA-treated adults with SZD suffering from pre-diabetes (HbA1c between 37 and 47 mmol/mol) and a BMI $\geq 27 \, kg/m^2$, when compared with placebo. Additionally, that semaglutide improves the following secondary endpoints: weight, schizophrenic symptoms (Positive and Negative Symptom Scale 6, PANSS-6), QoL-related measures, patient-related outcomes data and the cardiometabolic risk factor profile.

### METHOD

### Design

The current trial is an investigator-initiated, double-blinded, parallel, superiority, multicentre RCT with a 1:1 randomisation. In total, we will include 154 patients recruited from two Danish regional mental health services. A CONSORT diagram is provided in figure 1. Data management is purely digital and

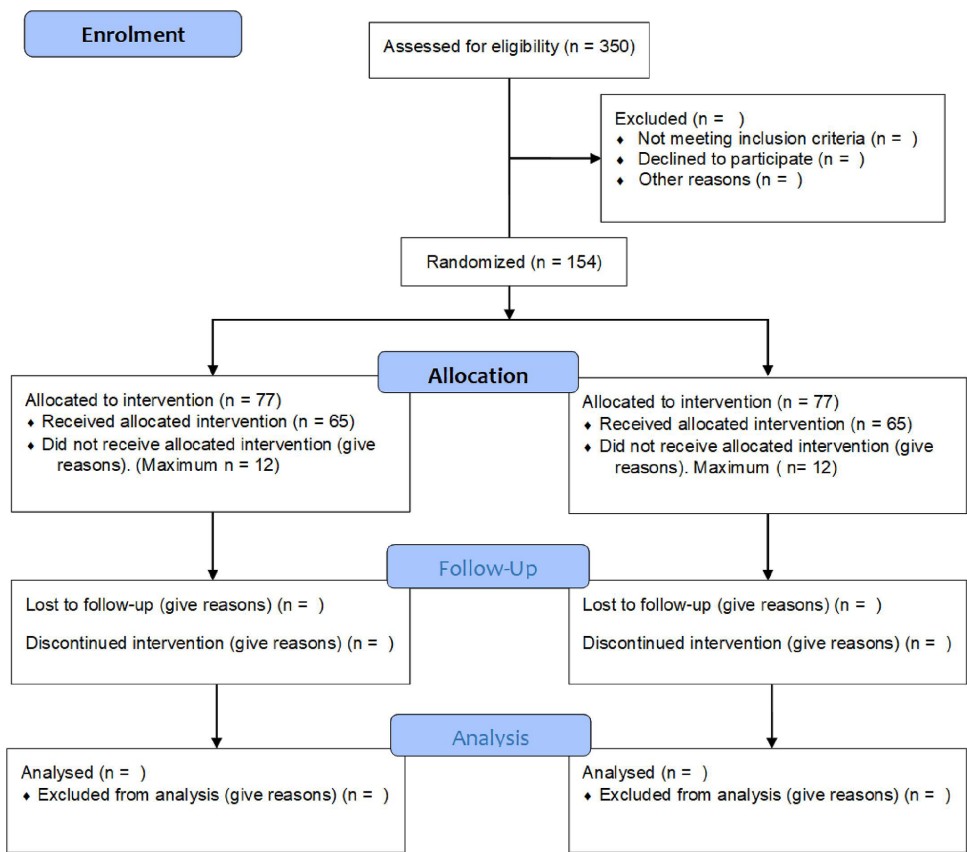

**Figure 1** Flow chart.

managed by Open Patient data Explorative Network, Odense University Hospital (OUH), Denmark, using REDCap.

## SETTING
The study will recruit patients from community mental health centres and private practitioners in the Region of Southern Denmark and the Region of Zealand, Denmark.

## PARTICIPANTS
Patients with a clinical diagnosis of either schizophrenia, schizotypal disorder or schizoaffective disorder (ICD-10 diagnoses: F20.x; F21; F25.x), age between 18 and 60 years receiving treatment in the regional mental healthcare of participating centres, or by private practice, and fulfilling criteria as listed in box 1.

## PROCEDURES
### Recruitment
Patients fulfilling inclusion criteria will be invited from community mental health centre clinics and private practices in the Region of Southern Denmark and the Region of Zealand. According to Danish clinical guidelines, patients treated with neuroleptics (almost exclusively SGAs) require minimum one annual laboratory test and clinical evaluation to test for the presence of the MetS.

Eligible subjects accepting to participate will be invited to screening visits and allocated to treatment with placebo or semaglutide at a dose of 1.0 mg administered subcutaneously once a week by blinded randomisation, if all inclusion criteria and none of the exclusion criteria are met. The PhD students will be in charge of recruitment participants, obtaining consent to participation, initiation of medication and execution of the trial including inclusion visits and follow-up visits.

### Randomisation and concealment
A total of 154 participants will be enrolled and randomised 1:1 to semaglutide or placebo. Block randomisation including blocks of either 4 or 6 randomisation numbers will be used, and the randomisations list will be provided by the manufacturer of semaglutide and placebo (Novo Nordisk A/S, Novo Allé, Bagsværd DK-2880 Denmark). Randomisation will be performed using REDCap.

### Mobile project nurses
The presence of schizophrenia impacts adherence and accordingly, adherence to study medication may be a challenge.[33] Therefore, we perform data collection at home by mobile study nurses to ensure optimal adherence and to minimise discomfort for the participants.

The study nurse will visit the subjects in their homes to administer injection and teach eligible subjects to self-inject under supervision. Thus, apart from the enrolment,

## Box 1

Inclusion criteria
⇒ Diagnosed with schizophrenia spectrum disorder (ICD-10 codes DF20, DF21 or DF25).
⇒ Age between 18 and 60 years (both included).
⇒ Approved contraception for female participants.
⇒ Treated by one of the community mental health centres and general practitioners in the Region of Southern Denmark or Zealand.
⇒ Antipsychotic SGA treatment for at least 6 months.
⇒ Stable comedication for at least 1 month.
⇒ HbA1c between 39 and 47 mmol/mol (both included). Two measurements with ≥3 month interval are required to confirm pre-diabetes. The first measurement is identified and obtained from patient journals, the second prior to enrolment.
⇒ Body mass index ≥27 kg/m$^2$. Two weights with ≥3 months interval are required to confirm obesity.
⇒ Capable of providing informed oral and written consent.
Exclusion criteria
⇒ Diagnosis of diabetes (type 1 diabetes (T1D) or T2D) or a HbA1c>47 mmol/mol.
⇒ Active malignant disease within the last 5 years.
⇒ Pregnancy or breast feeding.
⇒ A reported high consumption of alcohol or severe substance abuse.
⇒ Unwillingness to allow home visits by a study nurse.
⇒ Significant somatic disease: (1) end-stage renal failure (eGFR<15 mL/min); (2) elevated liver function tests (liver transaminases >2 times upper normal limit); (3) history of acute or chronic pancreatitis; (4) heart failure (NYHA class IV) or unstable angina pectoris or myocardial infarction with the last 6 months; (5) uncontrolled hypertension (systolic blood pressure >180 mm Hg, diastolic blood pressure >100 mm Hg).
⇒ Previous treatment with study drug or use of other weight reducing drugs within last 6 months.
⇒ Participation in other drug trials.
⇒ Treatment with drugs approved for overt diabetes type 2 except metformin.
⇒ Circumstances that the investigator believes will interfere with the trial.

we plan for a home-based study. If some patients are able to adhere to treatment and to self-inject, weekly visits may not be necessary. Thus, the frequency of home visits may be downscaled to involve monthly collection of data only. All data collection is allocated to study nurses and the PhD students, and the participants will maintain management-as-usual in the respective clinics from where they are recruited.

## Interventions

The intervention consists of 30 weeks of treatment with once-weekly subcutaneous injections of either semaglutide (1.34 mg/mL; 1.0 mg) or placebo. The current dose adjustment guidelines for Semaglutide will be followed (ie, starting at 0.25 mg weekly for 4 weeks, then 0.5 mg weekly for 4 weeks, and finally 1 mg or highest tolerated weekly dose until the end of intervention) or placebo (corresponding volume). Subjects, who develops side effects when reaching maximum dose, will be reduced

to the highest possible tolerated dose of semaglutide or placebo for the rest of the study.

Relevant concomitant care and interventions are permitted during the trial.

## Patient and public involvement

The project's design was developed in collaboration with individuals diagnosed with schizophrenia and their family members. Their valuable contributions have provided insights into the possibilities of conducting examinations within a home-based setting.

## Ethics and dissemination

This research has obtained approval from both the Danish Medicines Agency and The Regional Committees on Health Research Ethics for Southern Denmark. The results will be disseminated in peer review publications and conference presentations.

## OUTCOMES
### Primary outcome

Absolute reduction in HbA1c from baseline to end of treatment (30 weeks).

### Secondary outcomes

During the treatment and after 30 weeks of treatment:
a. Metabolic measures: BMI, blood pressure, triglyceride, cholesterol (HDL and LDL), waist circumference and HOMA-estimates based on fasting insulin and plasma glucose.
   Blood test will be taken at the nearest hospital prior to inclusion, at the inclusion, at 15 weeks and at end of study. BMI, waist circumference and blood pressure will be measured at weeks 1, 15 and 30.
b. PANSS-6: PANSS is an operationalised rating of positive and negative symptoms, in people with schizophrenia, which is sensitive to change.[34] PANSS-6 is a shorter version of the original 30-item version. It is validated and translated into Danish and has proven a good level of inter-rater reliability when rated based on the Simplified Negative and Positive Symptoms Interview (SNAPSI).[35 36] PANSS-6 interview will be performed at baseline, and after 15 and 30 weeks.
c. Short Form Survey (SF-36): SF-36 is a well-researched, self-reported measure of health. It comprises 36 questions in 8 different domains. The survey will be applied in Danish, using the latest updated version 2 (SF-36 V.2).[37] SF-36 V.2 will be measured at baseline, and after 15 and 30 weeks.
   Exploratory outcomes:
a. Cardiovascular risk markers: Cardiovascular autonomic neuropathy (CAN) (Vagus TM), and circulating levels of cardiovascular risk markers. CAN is considered a serious complication of diabetes, being associated with a fivefold increased risk of CVD.[38] Of interest to this study, CAN may be present prior to the development of diabetes[39] and it can be improved

by weight loss.[40] As CAN may be measured by a hand-held apparatus (Vagus TM), measurements are easily performed in the homes of the enrolled patients. CAN will be measured at inclusion, at 15 weeks and at end of study.

Moreover, the CAN results will be juxtaposed with various cardiometabolic risk markers, including proteins belonging to the GDF-family, activin, follistatin, the insulin-like growth factor-system and its regulating proteins and enzymes, and the adipokines leptin and adiponectin. All these data are explorative tertiary outcomes.

b. Proatherosclerotic changes: We want to perform a pilot study in a subsample of 20 participants from placebo vs 20 participants from the Semaglutide group, focusing on presymptomatic atherosclerosis assessed by positron emission tomography/CT (PET/CT) using 18F-sodium fluoride (NaF) as tracer. NaF-PET/CT is a novel way of detecting and measuring calcification in major arteries, potentially years or decades before manifest arterial wall calcification becomes detectable by cardiac CT.[41 42] We plan to perform two NaFPET/CT acquisitions, that is, at baseline and after 30 weeks treatment. Based on our experience, inclusion of 20 placebo-treated and 20 semaglutide-treated subjects is suitable to perform the pilot study. We hypothesise a ≥10% decrease in global disease score[43] as measured by NaF uptake in the heart and/or thoracic aorta, will be present after 30 weeks treatment.

c. Impact of Weight on Quality of Life-Lite (IWQOL-Lite): IWQOL-Lite is a reliable and valid self-report measure for assessing weight-related QoL in people with schizophrenia.[44] It provides a total score plus scores on five domains (physical function, self-esteem, sexual life, public distress and work). It is translated into 83 languages, including Danish. IWQOL-Lite will be measured at baseline, and after 15 and 30 weeks.

d. Medication Adherence Rating Scale (MARS): MARS is a validated and reliable measure of adherence for psychoactive medication. MARS constitutes 10 items rating the medical adherence.[45 46] MARS will be measured at baseline, and at weeks 5, 9, 15, 19, 23, 28 and 30.

e. The Simple Physical Activity Questionnaire (SIMPAQ): SIMPAQ has shown good test–retest reliability and has been validated against measures from Actigraph accelerometers.[47] SIMPAQ will be measured at baseline, and at weeks 5, 9, 15, 19, 23, 28 and 30.

f. The EuroQol five-dimensional (EQ-5D): EQ-5D questionnaire is a widely applied instrument for the measurement and cost–benefit evaluation of health interventions.[48] EQ-5D will be measured at baseline, and at weeks 5, 9, 15, 19, 23, 28 and 30.

A schedule of follow-up activities is shown in online supplemental table 1.

## SAFETY ASSESSMENTS

Adverse reactions will be evaluated weekly. At every 4-week visit, vital parameters will be recorded, including blood pressure, body weight and waist circumference. The safety parameters in the study include adverse reactions, vital parameters and liver and kidney function, which will be analysed at baseline, week 15 and week 30. If any side effects or changes in biochemical measures are observed, the trial physician or the physician responsible for the test will be informed.

All safety parameters will be recorded in an electronic case report form (eCRF) as soon as they have either been measured or reported by the study participant. The study will be monitored by Good Clinical Practice (GCP) and follow current GCP guidelines. The study protocol has been reviewed by GCP prior to inclusion of first participant, and subsequently, the study is monitored by GCP throughout the study period.

After drug administration, all potential side effects and other adverse events (AEs) observed will be reported in the eCRF. Semaglutide half-life is 1 week and thus expected to be completely eliminated after 6 weeks). Consequently, collection of AEs for all patients will be completed 6 weeks after the final injection by telephone calls. The nature of each AE will be assessed according to the severity and relationship of the experimental medicine or trial procedure and will be assessed by the primary investigator, who is a physician. If it is considered that the AE is related to the test drug or test procedure, it will be noted as a side effect in the eCRF. All severe AEs will be reported to the sponsor. Hospital admissions related to schizophrenia as soon as possible, other hospital admissions always within 24 hours of it coming to our attention. All severe adverse reactions (SARs) which, regardless of dose, result in death, are life-threatening, result in hospitalisation or prolongation of hospital stay in medical or surgical departments, result in significant or persistent disability or incapacity, or lead to congenital anomaly or malformation, will be considered in this trial to be unexpected (SUSAR).

Emergency unblinding will occur if the participants develop AEs that require knowledge of the treatment, if patient develop sign of overdose or if patient develop conditions that require treatment with the trial drug (semaglutide) or drugs with potential interactions with trial drug.

## STATISTICAL CONSIDERATIONS

Primary outcomes for the study will be absolute changes in HbA1c. Changes in HbA1c will be evaluated after 15 weeks and after 30 weeks.

Assuming that 30 weeks of semaglutide reduces HbA1c by 0.2%, and that SD equals 0.35%, 65 subjects in each arm are required to obtain a power of 90%, using a two-sided significance level of 5%.

Drop-out from trials concerning pharmacological treatment for obesity is non-linear; the drop-out from the treatment group will occur earlier due to side effects, while in the placebo group dropouts occur later due to lack of efficacy.

## Feasibility

In the SUSTAIN 1 trial (30 weeks of semaglutide in patients with T2D), the discontinuation of study drug amounted to 13% in those receiving 0.5 mg semaglutide, 12% in those receiving 1.0 mg semaglutide and 11% in placebo-treated subjects.[49] In the 16-week study with liraglutide in patients with schizophrenia, 23 of 214 patients declined to participate, and 111 patients did not meet inclusion criteria, leaving 103 to be enrolled. Of these, only seven patients dropped out of the study, leaving an overall success rate of 45%.[50] We believe to experience similar rates. Thus, we need to approach approximately 350 patients to enrol 154 patients. Given the number of potentially eligible patients in the two regions (currently estimated to >300 excluding new cases in the age 18–60 years), we believe this is feasible. In the present study, we will follow the participants for 30 weeks. Hence, we predict a higher risk of dropout. Accordingly, we conservatively assume a dropout rate of 15% to ensure that the required number of subjects to obtain power are recruited.

In summary, our study has been powered to examine a 0.2% decline in HbA1c, despite a drop-out rate of 15%, resulting in 77 patients in each arm. Initially, we considered implementation of an oral glucose tolerance test. However, after careful consideration, we concluded that such a test would not be feasible in a home-based setting. In seeking guidance, our user panel recommended against the oral glucose tolerance test, citing concerns about its practicality for patients. Taking this valuable input into account, we decided to refrain from incorporating the test, recognising the potential challenges it may pose for patients in performing such a procedure.

Although this study is not designed as treatment for obesity, we believe the estimation of drop-out is reliable as the study focus on the same patient population and the pharmacological profile of the investigated drug is similar to liraglutide.[50]

## DATA ANALYSIS

All participants randomised in the study will be included in the assessment of endpoints. All data will be analysed based on the intention-to-treat population. Clinical results will be presented as mean, least squares mean, least square mean change from baseline or least square difference between groups with SE, SD or two-sided 95% CI as appropriate. Statistical tests will be conducted as two-sided tests with a 5% significance level.

A mixed effects linear regression analysis will be used to compare changes in glucose tolerance, HbA1c, from baseline to 30 weeks follow-up between the intervention and the placebo groups. Missing values will be imputed, primary analysis will be intention to treat. Supplementary per-protocol analysis will accompany the intention-to-treat analysis.

All changes in secondary outcomes from baseline to various follow-up times are analysed using mixed effects linear regression analyses for continuous outcomes and mixed effects logistic regression analyses for categorical outcomes. All analyses on continuous outcomes are adjusted for the baseline value of the outcome, and in case of significant imbalances in baseline patient characteristics between the two randomised groups, exploratory analyses including such variables are conducted. To correct for multiple testing of the secondary outcomes, Holm's sequentially rejective multiple test procedure will be applied. Statistical analyses are performed using Stata software V. 16 (Statacorp). The hypothesis tests are two sided, and the 5% level of significance is considered.

## ETHICAL APPROVAL AND CONSENT TO PARTICIPATE

The project is approved by the Regional Science Ethical Committee for the Region of Southern Denmark and the Danish Medicine Agency. Data processing in connection with the study is reported to the region's list of ongoing research projects via the Executive Secretariat at Odense University Hospital (OUH). The study participants will not necessarily be able to benefit from participation in the study. Individual participants may be diagnosed with illness that may require treatment; in that case, they will be referred for relevant treatment. Participants will not have overt diabetes based on HbA1c. Therefore, following national guidelines, participants in this study are not candidates for standard treatment of diabetes and under normal circumstances they would not receive antidiabetic treatment or treatment that would reduce their risk of diabetes. However, metformin treatment of the MetS is accepted.

Amendments to the study protocol will be made to the Regional Science Ethical Committee for the Region of Southern Denmark and the Danish Medicine Agency.

The study will be conducted in accordance with the Declaration of Helsinki with amendments as well as in accordance with the General Data Protection Regulation, and the Danish Data Protection. In the case that biological material will be shipped to collaborators abroad, the Danish Protection Agency will be applied and Chapter V of the Danish Protection Act will be obeyed. It is mandatory that the clinical staff involved in the clinical trial has completed an official course in GCP.

## TRIAL STATUS

Recruitment opens January 2022 with first patient, first visit, whereas last patient, last visit (LPLV) is scheduled for September 2024. Analysis of primary endpoints is planned to March 2025: Data will be published in EudraCT within 12 months from LPLV. Publication of the final results on

primary endpoint is expected to be made public in March 2025.

## DISCUSSION

Metabolic disturbances, obesity and diabetes development in SGA treated patients are a major problem in the clinical setting, but the underlying mechanism is poorly understood, even though there are theories focusing on an interaction of medication, genes, life style, diet and physical activity.[8 13]

GLP-1 analogues have shown remarkable effect in the treatment of diabetes by contributing to weight loss, lowering blood glucose levels and improving mortality from CVD.[31 32 49] This is in contrast to life style interventions with physical activity and diet changes, as such initiatives have failed to induce robust long-term improvements of the metabolic disturbances.[22 23 51]

The main focus of this clinical trial is mitigation of metabolic aberrations due to SGA treatment and reversal of pre-diabetes. Hence, the primary outcome is the absolute reduction in HbA1c rather than weight loss.

Patients with severe mental illness experience many challenges due to schizophrenic symptoms and side effects to SGA treatment.[11 12] Recruitment may be difficult in this patient population, and accordingly, we are collaborating with local psychiatric teams in both regions and with local consultants to create a secure environment. The study is designed to focus on adherence to study protocol and minimise the patient's burden of study participation. The home visit by study nurses is likely to strengthen protocol adherence by assisting the participants and creating a secure and safe environment, eventually yielding higher reliability of data from interviews and questionnaires.

Semaglutide holds benefits compared with liraglutide when dealing with patients with weight gain caused by SGAs. Semaglutide requires weekly treatment only, which makes it easier to administer with help from a study nurse. In addition, the weight reducing effect of semaglutide is greater than liraglutide without compromising the cardioprotective effect.[52] Hence, if semaglutide 1.34 mg/mL has a beneficial effect on weight, HbA1c, MetS and QoL, it could potentially be used in patients with schizophrenia in high risk of CVD to prevent premature death due to diabetes.

Our study will attempt to mitigate the metabolic side effects of SGA induced overweight in individuals with schizophrenia and pre-diabetes. Treatment with semaglutide improves glycaemic control, weight and clinical cardiovascular risk factors by reducing numbers of fatal and non-fatal myocardial infarction and non-fatal stroke within 104 weeks.[53] We hypothesise that beneficial preclinical events are detectable by NAF uptake using PET-CT scans after 30 weeks of treatment. If so, this may contribute to a better understanding of the nature behind the increased cardiovascular risk within these patients.

We hope that our trial HISTORI will provide insight in how to benefit the cardiometabolic health and the QoL of schizophrenic patients—adding more and better years—and thereby change the HISTORY of this vulnerable group of patients. Additionally, if successful, HISTORI may pave the way for long-term, large-scale GLP-1 agonist interventions in schizophrenic patients with pre-diabetes, aiming to reduce the number of hardcore clinical cardiovascular endpoints, and thereby prolonging their life expectancy. Finally, a positive outcome may fuel initiatives to study the ability of GLP-1 agonists to prevent the development of the pre-diabetes, for example, by starting treatment shortly after treatment with SGAs are initiated in high-risk patients.

### Sponsor

Odense University Hospital

Steno Diabetes Centre Odense/Department of Endocrinology M. J.B. Winsløvvej 4. DK- 5000 Odense, Denmark.

### Primary investigator at the Region of Southern Denmark

Jan Frystyk, Clinical Professor, DMsc, Ph.D. Clinical management of patients with diabetes at SDCO; implementation of biomarkers in clinical studies, and PI on clinical studies, hereunder GCP-monitored RCT.

Department of Endocrinology M/Steno Diabetes Centre Odense (SDCO). Odense University Hospital. J.B. Winsløvvej 4. DK- 5000 Odense, Denmark.

### Primary investigator at the Region Zealand Community Psychiatry

Sidse Arnfred, Clinical Professor.

Specialist in family medicine, psychiatry and board certified supervisor in psychotherapy (CBT). Research expertise in schizophrenia, psychopathology and psychophysiology. Experienced PI on RCT's, qualitative studies of user experiences and treatment evaluation. Psychiatry West, Region Zealand, Research Unit West, Fælledvej 6, Building 3, 4. floor, DK-4200 Slagelse, Denmark.

### Primary investigator at the Region of Southern Denmark Community Psychiatry

Niels Bilenberg, Clinical Professor, Ph.D. OPUS clinics and community psychiatry centres of Region of Southern. Mental Health Hospital and University Clinic in the Region of Southern Denmark. J.B.Winsløws Vej 28 B, indg. 236, DK-5000 Odense, Denmark.

### Coinvestigators

Peter Gæde, MD, DMSci, Chief Physician, Associate Professor. Department of Steno Diabetes Centre Sjælland. Steno Diabetes Center Sjælland, Akacievej 7, 1.sal. DK-4300 Holbæk, Denmark.

### PhD students

Ashok Ganeshalingam, Physician, Ph.D. Student. Department of Endocrinology M/Steno Diabetes Centre Odense (SDCO). Odense University Hospital. J.B. Winsløvvej 4. DK-5000 Odense, Denmark.

Nicolai Gundtoft Uhrenholt, Physician, Ph.D. Student. Psychiatry West, Region Zealand, Research Unit West, Fælledvej 6, Building 3, 4. floor, DK-4200 Slagelse, Denmark.

**Author affiliations**
¹Endocrine Research Unit, Department of Endocrinology, Odense University Hospital & Department of Clinical Research, Faculty of Health, Odense Universitetshospital, Odense, Denmark
²Department of Internal Medicine, Lillebaelt Hospital - University Hospital of Southern Denmark, Kolding, Denmark
³Department of Endocrinology, Odense University Hospital, Odense, Denmark
⁴Psychiatry West, Region Zealand, Research Unit West, Slagelse, Denmark, Slagelse, Denmark
⁵Department of Child and Adolescent Mental Health Odense, Mental Health Services, University of Southern Denmark, Odense, Denmark
⁶University of Copenhagen, Kobenhavn, Denmark
⁷Department of Cardiology and Endocrinology, Slagelse Hospital, Slagelse, Denmark
⁸Department of Clinical Research, Faculty of Health Sciences, University of Southern Denmark, Odense, Denmark

**Contributors** AAG, NGU, SA, PHG, NB and JF initiated the study design, AAG and NGU helped with implementation and execution of the study. JF, NB, SA and PHG are grant holders. JF and NB provided statistical expertise in clinical trial design, AAG and NGU are conducting the primary statistical analysis. AAG made significant contributions to the study protocol in the design and the writing process in with contributions from NGU. All authors contributed to refinement of the study protocol and approved the final manuscript. Data from this study belong to the research group behind Project HISTORI and can be accessed after application to the sponsor JF.

**Funding** Novo Nordisk Foundation: DKK8 584 959 (salary and tuition fee PHD Students and study-nurses, running expenses, biochemical and laboratory procedures, TAP personal, mileage allowance). Research Grant from Steno Diabetes Center Zealand, Denmark: DKK660 000. Research Grant from Steno Diabetes Center Odense, Denmark: DKK578 000. Annum (Region Zealand) to PhD-student Nicolai Uhrenholt: DDK20 000 per year for 3 years (DKK60 000 in total). Slagelse Puljen 2020: The Region of Zealand, Denmark: DKK240 000. Slagelse Puljen 2022: DKK150 000 DKK. Region Sjællands Sundhedsvidenskabelige Forskningsfond 2022: DKK252 631. Furthermore, Novo Nordisk A/S will provide investigational drug and placebo free of charge for sponsor, but is otherwise not involved in the study. The scientists involved will conduct the experiment out of general scientific interest without personal financial gain. None of the investigators involved have financial interests (including shares, direct employment, members of advisory boards) in the drug company that produces the active drug used in the study. None of the participants in the study will receive any provision for participation. Participants will receive mileage allowances. Any harm as a result of study participation will be covered by the Danish Patient Compensation.

**Competing interests** None declared.

**Patient and public involvement** Patients and/or the public were involved in the design, or conduct, or reporting, or dissemination plans of this research. Refer to the Methods section for further details.

**Patient consent for publication** Not applicable.

**Provenance and peer review** Not commissioned; externally peer reviewed.

**ORCID iD**
Ashok Ainkaran Ganeshalingam http://orcid.org/0000-0003-2218-5045

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
