## [Reviewer comments · BMJ Open]

ARTICLE DETAILS

TITLE (PROVISIONAL)	Home based Intervention with Semaglutide Treatment of Neuroleptic-Related Prediabetes (HISTORI): protocol describing a prospective, randomized, placebo controlled and double blinded multicenter trial
AUTHORS	Ganeshalingam, Ashok; Uhrenholt, Nicolai; Arnfred, Sidse; Gæde, Peter; Bilenberg, Niels; Frystyk, Jan

VERSION 1 – REVIEW

REVIEWER	Malik, Rayaz A. Weill Cornell Medicine-Qatar, Medicine
REVIEW RETURNED	04-Oct-2023

GENERAL COMMENTS	This protocol is well written for an important study. The assessment of CAN is important but seems almost like an afterthought. Is the proposed study powered to show an effect on CAN, especially in 30 weeks when 8 weeks will be used to titrate to 1.0mg Semaglutide. Page 11- GCP is good clinical practice, not good clinical praxis. Where has the 0.2% drop in HbA1c been derived from? I would expect there to be a larger decrease. How is the weight going to be assessed at home? Will there be any assessment on appetite and smoking as GLP-1 agonists impact on eating and smoking behaviour.
---

REVIEWER	Muzurović, Emir M. University of Montenegro
REVIEW RETURNED	27-Nov-2023

GENERAL COMMENTS	Ganeshalingam et al. designed the trial with the aim is to investigate whether 30 weeks add -on treatment with the GLP-1 RA semaglutide can reduce HbA1c sufficiently to reverse prediabetes and the metabolic syndrome (MetS) in overweight schizophrenic patients. This trial is well/designed. Minor comments: Introduction need to be shortened a bit with the aim to be more concise. In the introduction it is very important to publish recently published paper on the SELECT trial, and recently published Editorial in the JDC: / Lincoff AM, Brown-Frandsen K, Colhoun HM, Deanfield J, Emerson SS, Esbjerg S, Hardt-Lindberg S, Hovingh GK, Kahn SE, Kushner RF, Lingvay I, Oral TK, Michelsen MM, Plutzky J, Tornøe
--

	CW, Ryan DH; SELECT Trial Investigators. Semaglutide and Cardiovascular Outcomes in Obesity without Diabetes. N Engl J Med. 2023 Nov 11. doi: 10.1056/NEJMoa2307563. - Muzurović E, Yumuk VD, Rizzo M. GLP-1 and dual GIP/GLP-1 receptor agonists in overweight/obese patients for atherosclerotic cardiovascular disease prevention: Where are we now? J Diabetes Complications. 2023 Nov 7;37(12):108647. doi: 10.1016/j.jdiacomp.2023.108647. Epub ahead of print. PMID: 37952274.
--	---

REVIEWER	Obermayer, Anna Medical University of Graz
REVIEW RETURNED	19-Jan-2024

GENERAL COMMENTS	This protocol of a randomized, placebo controlled and double blinded multicenter trial is well written, clear and the methods are described sufficiently to allow the study to be repeated. I am looking forward to the results. What seems to be missing from this protocol would be assessments of the patients eating habits and changes over time (such as a food frequency questionnaire, a food diary, an app to record food or similar tools) because changes in the eating behaviour might be a confounding factor. Furthermore the seasonal changes should be considered as well because christmas, easter and other holidays might change the eating behaviour significantly. Additionally any infections and the possible subsequent blood sugar elevations need to be recorded. HbA1c allows an overview of the last 3 months but episodes of hypoglycaemia need to be recorded in order to ensure any improvement in HbA1c is not achieved by hypoglycaemic episodes which might be detrimental for the participant. Furthermore weight loss should be specified into fat loss, muscle loss and bone loss, because not all weight lost is equal. An oral glucose tolerance test would also be more informative than just fasting blood tests. As the weight lost with GLP-1-RA appears to be regained over time after injections are stopped, are there plans to provide patients with further medication after the end of the study and are patients in the control group able to access the medication at the end of the study? While there are several limitations, overall this study will provide important information on the use of GLP-1-RA in schizophrenia.
--

VERSION 1 – AUTHOR RESPONSE

Reviewer: 1

Dr. Rayaz A. Malik, Weill Cornell Medicine-Qatar

Comments to the Author:

This protocol is well written for an important study.

The assessment of CAN is important but seems almost like an afterthought.

Is the proposed study powered to show an effect on CAN, especially in 30 weeks when 8 weeks will be used to titrate to 1.0mg Semaglutide.

Page 11- GCP is good clinical practice, not good clinical praxis.

Where has the 0.2% drop in HbA1c been derived from? I would expect there to be a larger decrease. How is the weight going to be assessed at home?

Will there be any assessment on appetite and smoking as GLP-1 agonists impact on eating and smoking behaviour.

Dear Dr. Rayaz,

Thank you for dedicating your time and providing valuable input to our study protocol. We deeply appreciate all your insights. Your positive feedback on our protocol holds great significance for us.

1. The assessment of CAN is important but seems almost like an afterthought.

Is the proposed study powered to show an effect on CAN, especially in 30 weeks when 8 weeks will be used to titrate to 1.0mg Semaglutide.

We actually know the person, who developed the Vagus® apparatus used to measure cardiovascular autonomic neuropathy (CAN). Therefore, this variable was indeed included when we outlined the protocol. However, we fully admit that the CAN measurement was included as an explorative variable. A later study (Hansen et al., Frontiers 2019) showed that following 24 weeks of liraglutide treatment in obese type 1 diabetic patients, there was a significant weight loss, but no changes in CAN. Thus, we are keen to see what this study shows, considering that semaglutide is more potent than liraglutide and that we treat for 30 and not 24 weeks.

We aim to utilize CAN to evaluate any positive response to a 30-week treatment with Semaglutide in comparison to placebo treatment. The CAN results will be compared against changes in HbA1c and weight, providing hypothesis generating rather than causal data.

Moreover, the CAN results will be juxtaposed with various cardio-metabolic risk markers, including proteins belonging to the GDF-family, Activin, Follistatin, the IGF-system and its regulating proteins and enzymes, and the adipokines leptin and adiponectin. All these data are explorative tertiary outcomes. We have stressed the in the manuscript: Tertiary exploratory outcomes include...(lines 224-234).

In the context of explorative data, we collect blood samples at baseline and week 30 for bio-banking and subsequent studies. This comprehensive approach is anticipated to offer novel insights into the potential cardiovascular protective effects of GLP-1Ra.

It's important to note that the study is not specifically powered to demonstrate an effect on CAN; rather, its power is directed towards showing an effect on HbA1c reduction, highlighting the prioritized focus on this particular aspect of the research.

Page 11- GCP is good clinical practice, not good clinical praxis.

Thank you for noticing. We will correct this spelling mistake.

Where has the 0.2% drop in HbA1c been derived from? I would expect there to be a larger decrease.

We have conducted a thorough review of previous studies involving GLP-1Ra in patients with schizophrenia, revealing a consensus that a 0.2% decline in HbA1c is considered a significant reduction. Thus, our study was powered according to this decline in HbA1c, adding a drop-out rate of 15%.

Initially, we were considering implementation of an oral glucose tolerance test. However, after careful consideration, we concluded that such a test would not be feasible in a home-based setting.

In seeking guidance, our user panel recommended against the oral glucose tolerance test, citing concerns about its practicality for patients. Taking this valuable input into account we decided to refrain from incorporating the test, recognizing the potential challenges it may pose for patients in performing such a procedure.

How is the weight going to be assessed at home?

The research personnel will transport all necessary equipment to participants' homes during each home visit. Measurements will be consistently taken on the same scale at the same location within their homes on each occasion. This has been added to the text; line 185-186..

Will there be any assessment on appetite and smoking as GLP-1 agonists impact on eating and smoking behaviour.

Thank you for highlighting the absence of assessments related to patients' eating habits. We initially considered the Yale Food Addiction Scale (YFAS) but opted for other questionnaires due to resource constraints and alignment with existing evidence from prior Randomized Controlled Trials (RCTs) in patients with schizophrenia. Although we cannot modify the current study, your insights inspire us to reassess the feasibility of incorporating additional assessments in future studies, including tools like a food frequency questionnaire, a food diary, or an app to record food. Your feedback is valued as we continually strive to enhance the comprehensiveness of our protocols.

Reviewer: 2

Dr. Emir M. Muzurović, University of Montenegro

Comments to the Author:

Ganeshalingam et al. designed the trial with the aim is to investigate whether 30 weeks add -on treatment with the GLP-1 RA semaglutide can reduce HbA1c sufficiently to reverse prediabetes and the metabolic syndrome (MetS) in overweight schizophrenic patients.

This trial is well/designed.

Minor comments:

Introduction need to be shortened a bit with the aim to be more concise.

In the introduction it is very important to publish recently published paper on the SELECT trial, and recently published Editorial in the JDC:

/ Lincoff AM, Brown-Frandsen K, Colhoun HM, Deanfield J, Emerson SS, Esbjerg S, Hardt-Lindberg S, Hovingh GK, Kahn SE, Kushner RF, Lingvay I, Oral TK, Michelsen MM, Plutzky J, Tornøe CW, Ryan DH; SELECT Trial Investigators. Semaglutide and Cardiovascular Outcomes in Obesity without Diabetes. *N Engl J Med.* 2023 Nov 11. doi: 10.1056/NEJMoa2307563.

- Muzurović E, Yumuk VD, Rizzo M. GLP-1 and dual GIP/GLP-1 receptor agonists in overweight/obese patients for atherosclerotic cardiovascular disease prevention: Where are we now? *J Diabetes Complications.* 2023 Nov 7;37(12):108647. doi: 10.1016/j.jdiacomp.2023.108647. Epub ahead of print. PMID: 37952274.

Dear Dr. Emir M. Muzurović,

Thank you for dedicating your time and providing valuable input to our study protocol. We deeply appreciate all your insights. Your positive feedback on our protocol holds great significance for us.

Introduction need to be shortened a bit with the aim to be more concise.

We have streamlined the introduction section to encompass both psychiatric and multiple metabolic endpoints in our study. The instruction has been shortened please see the introduction section.

Thank you for noting the omission of recent publications on the SELECT trial in our study protocol. Our trial protocol was submitted in mid-July 2023, predating the publications you mentioned. But we appreciate your input and have thoroughly examined the perspectives of the studies you referenced. They are now included in the introduction section..

Reviewer: 3

Dr. Anna Obermayer, Medical University of Graz

Comments to the Author:

This protocol of a randomized, placebo controlled and double blinded multicenter trial is well written, clear and the methods are described sufficiently to allow the study to be repeated. I am looking forward to the results.

What seems to be missing from this protocol would be assessments of the patients eating habits and changes over time (such as a food frequency questionnaire, a food diary, an app to record food or similar tools) because changes in the eating behaviour might be a confounding factor. Furthermore the seasonal changes should be considered as well because Christmas, Easter and other holidays might change the eating behaviour significantly.

Additionally any infections and the possible subsequent blood sugar elevations need to be recorded.

HbA1c allows an overview of the last 3 months but episodes of hypoglycaemia need to be recorded in order to ensure any improvement in HbA1c is not achieved by hypoglycaemic episodes which might be detrimental for the participant.

Furthermore weight loss should be specified into fat loss, muscle loss and bone loss, because not all weight lost is equal.

An oral glucose tolerance test would also be more informative than just fasting blood tests.

As the weight lost with GLP-1-RA appears to be regained over time after injections are stopped, are there plans to provide patients with further medication after the end of the study and are patients in the control group able to access the medication at the end of the study?

While there are several limitations, overall this study will provide important information on the use of GLP-1-RA in schizophrenia.

Dear Dr. Anna Obermayer,

Thank you for dedicating your time and providing valuable input to our study protocol. We deeply appreciate all your insights. Your positive feedback on our protocol holds great significance for us.

Regarding eating habits:

We genuinely appreciate your keen observation regarding the absence of assessments related to patients' eating habits and potential changes over time. Your suggestion of incorporating tools such as a food frequency questionnaire, a food diary, or an app to record food is indeed valuable, as changes in eating behavior can pose as confounding factors.

Indeed, we did consider the Yale Food Addiction Scale (YFAS) when initially conceptualizing the study. However, due to resource constraints and the need to prioritize based on existing evidence, we opted for other questionnaires, which were more aligned with the available data at the time. Specifically, our decision was influenced by the findings from other Randomized Controlled Trials (RCTs) conducted in patients with schizophrenia.

Your valuable insights have motivated us to reassess the feasibility of including additional assessments. While we are unable to modify the current study design, we commit to integrating assessments such as YFAS into our future studies as part of our ongoing efforts to enhance the comprehensiveness of our protocols.”

Regarding safety and hypoglycemia:

We prioritize the vigilant monitoring of potential side effects, particularly hypoglycemia, in our study. Adverse events are systematically recorded using dedicated report forms. Participants undergo weekly evaluations for the initial 8 weeks and then biweekly assessments to promptly identify any potential side effects, including hypoglycemia.

Currently, the majority of participants have completed the Lead-in Placebo Lead-out Visit (LPLV). Notably, only one participant exhibited a blood glucose level below 4.0, and none experienced levels below 3.0. It is essential to highlight that GLP-1Ra are infrequently associated with hypoglycemia. In the rare instances reported in clinical settings, patients typically received other glucose-lowering drugs concurrently.

Regarding OGTT:

Initially, we were contemplating the implementation of an oral glucose tolerance test. However, after careful consideration, we concluded that such a test would not be feasible in a home-based setting.

In seeking guidance, our user panel recommended against the oral glucose tolerance test, citing concerns about its practicality for patients. Taking this valuable input into account we decided to refrain from incorporating the test, recognizing the potential challenges it may pose for patients in performing such a procedure.

Regarding Weight loss:

Thank you for highlighting a limitation in our study. Due to the nature of our home-based intervention, tailored for individuals with severe symptoms of schizophrenia and involving minimal hospital visits, we did not conduct full body Dual-Energy X-ray Absorptiometry (DXA) scans to assess body compositions.

To address this gap, we have initiated a pilot study utilizing Sodium Fluoride Positron Emission Tomography Computed Tomography (PET CT) scans on 40 participants, ideally split evenly between the active treatment and placebo groups. While the primary focus of these scans is to assess

preclinical atherosclerotic changes and the potential effects of GLP-1Ra in mitigating these changes, they also offer an opportunity to evaluate body composition. This additional analysis will provide valuable insights into the distribution of weight loss among participants.

Regarding Future:

We acknowledge current studies indicating weight regain upon medication cessation, particularly in obese individuals with prolonged obesity, often unrelated to other drug treatments. Many participants in our study experienced weight gain due to Second-Generation Antipsychotic (SGA) treatment. Our hypothesis and hope is that some participants may retain the achieved weight loss.

Since the protocol approval, the weight losing drug Wegovy from Novo Nordisk containing 2.4 mg Semaglutide weekly, was introduced. Unfortunately, this treatment is expensive, making it unaffordable for our participants.

Recently, we obtained approval from the Regional Ethical Committee of the Region of Southern Denmark for a 1-year follow-up on participants from this region. This follow-up aims to evaluate whether participants are maintaining the achieved weight loss. Currently, 85 participants have agreed to participate in the 1-year follow-up, and we eagerly anticipate uncovering insights into this question.

Reviewer: 1

Competing interests of Reviewer: none

Reviewer: 2

Competing interests of Reviewer: No Col

Reviewer: 3

Competing interests of Reviewer: none

VERSION 2 – REVIEW

REVIEWER	Obermayer, Anna Medical University of Graz
REVIEW RETURNED	18-Feb-2024
GENERAL COMMENTS	Thank you for the revised manuscript. I am pleased to accept it for publication.